# Mutual potentiation drives synergy between trimethoprim and sulfamethoxazole

Yusuke Minato [1], Surendra Dawadi[2], Shannon L. Kordus[1], Abiram Sivanandam[1], Courtney C. Aldrich[2] & Anthony D. Baughn [1]

Trimethoprim (TMP)-sulfamethoxazole (SMX) is a widely used synergistic antimicrobial combination to treat a variety of bacterial and certain fungal infections. These drugs act by targeting sequential steps in the biosynthetic pathway for tetrahydrofolate (THF), where SMX inhibits production of the THF precursor dihydropteroate, and TMP inhibits conversion of dihydrofolate (DHF) to THF. Consequently, SMX potentiates TMP by limiting de novo DHF production and this mono-potentiation mechanism is the current explanation for their synergistic action. Here, we demonstrate that this model is insufficient to explain the potent synergy of TMP-SMX. Using genetic and biochemical approaches, we characterize a metabolic feedback loop in which THF is critical for production of the folate precursor dihydropterin pyrophosphate (DHPPP). We reveal that TMP potentiates SMX activity through inhibition of DHPPP synthesis. Our study demonstrates that the TMP-SMX synergy is driven by mutual potentiation of the action of each drug on the other.

---

[1] Department of Microbiology and Immunology, University of Minnesota Medical School, 689 23rd Avenue SE, Minneapolis, Minnesota 55455, USA. [2] Department of Medicinal Chemistry, University of Minnesota, 689 23rd Avenue SE, Minneapolis, Minnesota 55455, USA. These authors contributed equally: Surendra Dawadi, Shannon L. Kordus and Abiram Sivanandam. Correspondence and requests for materials should be addressed to Y.M. (email: yminato@umn.edu) or to A.D.B. (email: abaughn@umn.edu)

Some antimicrobial drug combinations show strongly synergistic effects, where the combined inhibitory activity is far greater than the sum of individual activities[1]. However, in most cases, it is not clear why combinations may act synergistically. A mixture of trimethoprim (TMP) and sulfamethoxazole (SMX), also known as Co-trimoxazole, is a widely used synergistic antimicrobial combination to treat a variety of bacterial infections[2]. TMP-SMX is also effective against certain fungal infections and is the major treatment choice for pneumocystis pneumonia, which is one of the most common opportunistic infections in people with HIV-AIDS[3]. In bacteria, SMX inhibits dihydropteroate (DHPte) production from the two folate precursors, *p*-aminobenzoic acid (PABA) and 6-hydroxymethyl-7,8-dihydropterin pyrophosphate (DHPPP) (Fig. 1a)[4]. TMP inhibits the reduction of dihydrofolate (DHF) to tetrahydrofolate (THF)[5]. Although no conclusive evidence has been provided, it is generally accepted that the potent synergy between these drugs derives simply from the sequential inhibition of adjacent steps in microbial THF biosynthesis[6]. Mathematical modeling of this pathway suggests that this model is sufficient to explain how SMX action results in potentiation of TMP action, yet is inadequate to explain how TMP action potentiates SMX action[7].

In this study, we take a systematic genetic approach, using *Escherichia coli* single-gene deletion mutants[8], and discover that inhibition of DHPPP biosynthesis increases SMX activity. We also identify a functional metabolic feedback loop in the folate biosynthesis pathway by which TMP can also limit DHPPP biosynthesis. Collectively, our study indicates that TMP also potentiates SMX activity, and that the strong synergy between SMX and TMP is mediated by mutual potentiation. Our findings reveal a novel mechanism of drug synergism underlying the therapeutic efficacy of a widely established combination antimicrobial treatment and suggest that other metabolic pathways with functional feedback loops might be similarly susceptible to synergistic inhibitors.

## Results

**Inhibition of sequential steps in THF synthesis is not always synergistic.** To test whether inhibition of other sequential steps in the THF pathway can produce synergistic activity we targeted synthesis of PABA, an essential precursor for DHPte, with the antimicrobial compound 3,3-dichloro-1-(3-nitrophenyl) prop-2-en-1-one (MAC173979), which has been shown to inhibit PABA synthesis in *E. coli*[9] (Fig. 1a). As this biosynthetic module is located upstream of the SMX target, MAC173979 is expected to be synergistic with both SMX and TMP. We performed checkerboard assays to determine fractional inhibitory concentration indexes (FICIs) as a measure of interaction between antimicrobial agents[10]. We found that the combination of MAC173979 and SMX was only mildly synergistic (Fig. 1b and Supplementary Fig. 1a) and the combination of MAC173979 and TMP was merely additive and not synergistic (Fig. 1c and Supplementary Fig. 1b), while consistent with there being an unexplained aspect of synergy between TMP and SMX (Fig. 1d and Supplementary Fig. 1c). Further, although MAC173979 could potentiate susceptibility to SMX and, to a lesser degree, TMP, SMX and TMP did not potentiate susceptibility to MAC173979 (Fig. 1b,c). Thus, the interaction of the PABA biosynthesis inhibitor with both SMX and TMP is one of mono-potentiation. These FICI profiles show a stark contrast with the strong mutual potentiation observed between SMX and TMP (Fig. 1d and Supplementary Fig. 1c). Similar FICI profiles were observed for a clinical isolate of *E. coli* (strain B11)[11] and methicillin-resistant *Staphylococcus aureus* (strain USA300)[12], demonstrating that MAC173979 and TMP were not synergistic, whereas SMX and TMP were synergistic against these strains (Table 1). These results confirmed that a model based on sequential inhibition within the bacterial THF biosynthesis pathway is not adequate to explain the potent synergy between SMX and TMP.

**Folate precursors affect susceptibility to SMX and TMP.** To further evaluate synergy through targeting of sequential steps in

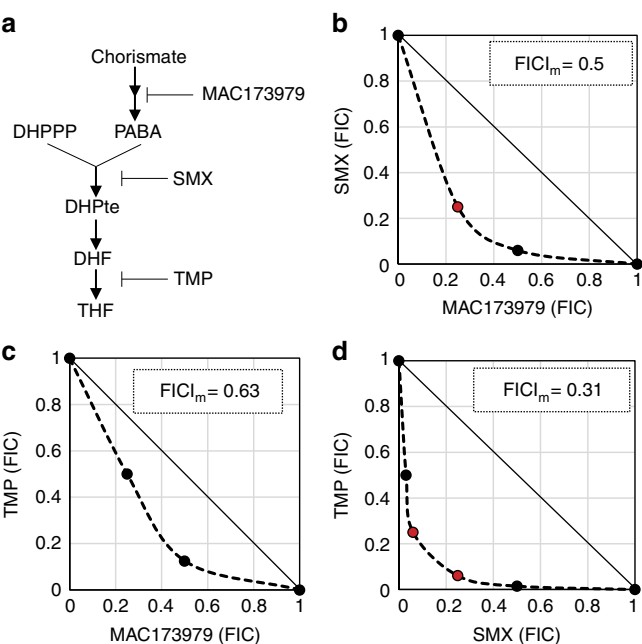

**Fig. 1** Synergistic activity of anti-folate combinations against *E. coli* and *S. aureus*. **a** Targets of anti-folate compounds. **b–d** *E. coli* BW25113 strain was grown overnight in LB medium. Cultures were washed twice and resuspended in M9-glucose, then inoculated into a 96-well round-bottom plate (Corning) containing the same medium with a range of concentrations of SMX, TMP, MAC173979, or combination of two compounds. Concentration ranges were as follows: SMX (0.024–25 μg ml$^{-1}$), TMP (0.0078–1 μg ml$^{-1}$), and MAC173979 (0.05–25 μg ml$^{-1}$). MICs were determined by visible growth after 24 h incubation at 37 °C. Synergy was assessed by calculating FICI. FICI$_m$, minimum value of FICI in the tested combinations is shown. Synergy (FICI$_m$ ≤ 0.5). No interaction (FICI$_m$ > 0.5). **b–d** Graphical representations of *E. coli* BW25113 checkerboard assays are shown. Representative data from at least three independent experiments are shown. **b** SMX and MAC173979. **c** TMP and MAC173979. **d** SMX and TMP

**Table 1** MICs of SMX, TMP, and MAC173979 and FICI$_m$ of SMX-TMP and MAC173979-TMP

| Bacterial strains | MIC (μg ml$^{-1}$) | | | FICI$_m$ | |
| --- | --- | --- | --- | --- | --- |
| | SMX | TMP | MAC173979 | SMX-TMP | MAC173979-TMP |
| *E. coli* BW25113 | 1.6 | 0.60 | 1.6 | 0.31 | 0.63 |
| *E. coli* B11 | 3.1 | 1.0 | 0.80 | 0.31 | 1.0–2.0 |
| *S. aureus* USA300 | 0.80 | 0.50 | 6.3 | 0.16 | 1.0 |

FICI$_m$, minimum fractional inhibitory combination of antimicrobial agent pairs found to achieve growth inhibition; MIC, minimum concentration of antimicrobial agent required to inhibit at least 50% of growth relative to a no drug control after 24 h of incubation at 37 °C; SMX, sulfamethoxazole; TMP, trimethoprim

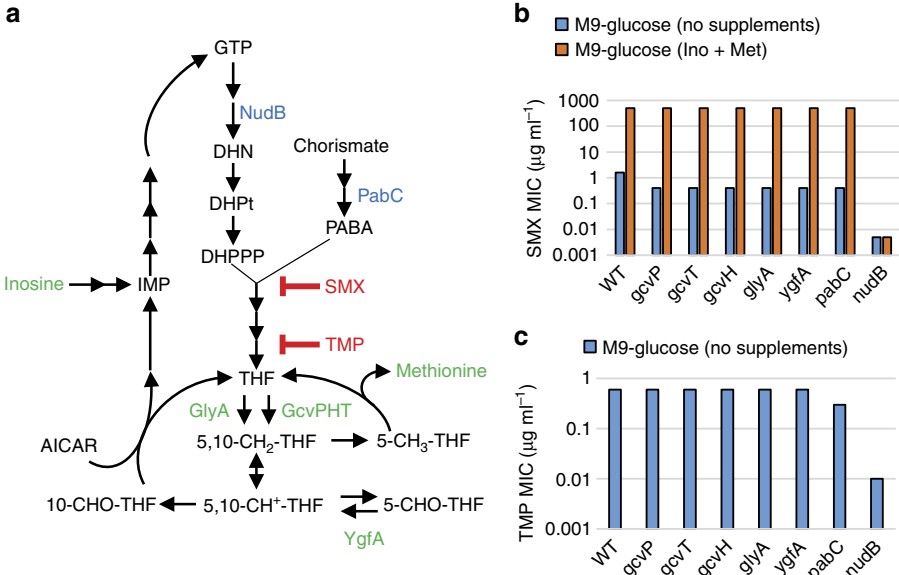

**Fig. 2** Folate deficiency potentiates SMX activity. **a** Schematic of *E. coli* folate metabolism. Green characters indicate metabolites and enzymes that affect SMX susceptibility. Blue characters indicate enzymes that affect both SMX and TMP susceptibility. **b** MICs of SMX and TMP for *E. coli* folate pathway mutants were determined after 24 h of incubation at 37 °C in M9-glucose medium. The selected *E. coli* single-gene deletion mutants and their parental strain BW25113 (WT) from the Keio collection[8] were used. **c** Effects of inosine (Ino) and methionine (Met) on SMX MIC against *E. coli* BW25113. Ino and Met were added to the medium at 10 µg ml$^{-1}$. MICs of SMX were determined after 24 h incubation at 37 °C in M9-glucose medium. Representative data from at least three independent experiments are shown

THF synthesis, we assessed the impact of genetic impairment of steps upstream of DHPte synthesis on potency of SMX and TMP (Fig. 2a). It was previously suggested[13] and recently demonstrated that SMX acts by competing with PABA for ligation with DHPPP[14]. As a result, SMX forms dead-end complexes with DHPPP (dihydropterin-SMX)[15] and inhibits DHPte production through metabolic wasting[13, 16]. Based on this model of metabolic wasting, we expected SMX activity would be influenced by the intracellular abundance of both PABA and DHPPP (Fig. 2a). We previously found that genetic disruption of the PABA biosynthesis pathway potentiates SMX activity against *Mycobacterium tuberculosis*[17]. Similar to *M. tuberculosis*, an *E. coli* mutant strain deleted for *pabC*, encoding the aminodeoxychorismate lyase involved in PABA biosynthesis, showed 4-fold enhanced susceptibility to SMX (Fig. 2b). In contrast, the *E. coli pabC* deletion mutant strain was only twofold more susceptible to TMP (Fig. 2c). This observation is consistent with the finding that the combination of MAC173979 and SMX is mildly synergistic (Fig. 1b and Supplementary Fig. 1a), whereas the combination of MAC173979 and TMP is merely additive (Fig. 1c and Supplementary Fig. 1b). To determine the impact of DHPPP levels on SMX and TMP susceptibility, we utilized an *E. coli* mutant strain deleted for *nudB*, which encodes dihydroneopterin triphosphate pyrophosphohydrolase[18]. Interestingly, this strain showed over 300-fold enhanced susceptibility to SMX and 60-fold enhanced susceptibility to TMP (Fig. 2b, c), demonstrating that DHPPP levels are far more impactful than PABA levels on SMX and TMP susceptibility. Of note, the Δ*nudB* strain showed the same susceptibility to ceftazidime and ciprofloxacin as the parent strain, indicating that the enhanced drug susceptibility phenotype of the Δ*nudB* strain was specific to anti-folate drugs (Supplementary Table 1).

**Folate-dependent metabolites affect susceptibility to SMX but not TMP.** As DHPPP is ultimately derived from the folate-

dependent purine nucleotide guanosine-5'-triphosphate (GTP), we were curious to explore whether alterations in folate interconversion could impact on the susceptibility of *E. coli* to TMP and SMX. The genes *glyA* and *gcvPHT* code for serine hydroxymethyltransferase and the glycine cleavage system, respectively. These two independent steps yield the 10-formyl-THF precursor 5,10-methylene-THF from THF (Fig. 2a) and were previously associated with susceptibility to anti-folate drugs in a high throughput conditional gene essentiality study[19, 20]. We observed that the *E. coli* single-gene deletion mutants in these two steps had enhanced susceptibility to SMX (Fig. 2b) and complementation of individual mutants completely restored SMX susceptibility to the level of the parental strain (Supplementary Table 2). Interestingly, the *glyA* and *gcv* deletion mutant strains showed a level of TMP susceptibility that was comparable to that of the parental strain (Fig. 2c). These observations demonstrate that interference with 10-formyl-THF production mediates enhanced susceptibility to SMX and suggests that disruption of folate-dependent metabolism is the likely basis for potentiation of SMX activity by TMP.

We next investigated how disruption of 10-formyl-THF synthesis impacts SMX susceptibility. It has long been known that the combination of two THF-dependent metabolites, inosine and methionine, antagonizes SMX activity by an, as yet, uncharacterized mechanism (Fig. 2b)[21]. We found that these metabolites also antagonized SMX susceptibility in the *E. coli glyA* and *gcv* deletion mutant strains. Since these deletion mutants showed the same level of SMX susceptibility as the parental strain in the presence of inosine and methionine, they are able to overcome the limited availability of 10-formyl-THF by supplementation with these two THF-dependent metabolites. Of note, inosine and methionine had no effect on TMP susceptibility in *E. coli* (Supplementary Fig. 2). Given that 10-formyl-THF is the essential cofactor for the de novo purine biosynthesis pathway and biosynthesis of the folate precursor DHPPP starts from GTP[4], we hypothesized that limitation of 10-formyl-THF

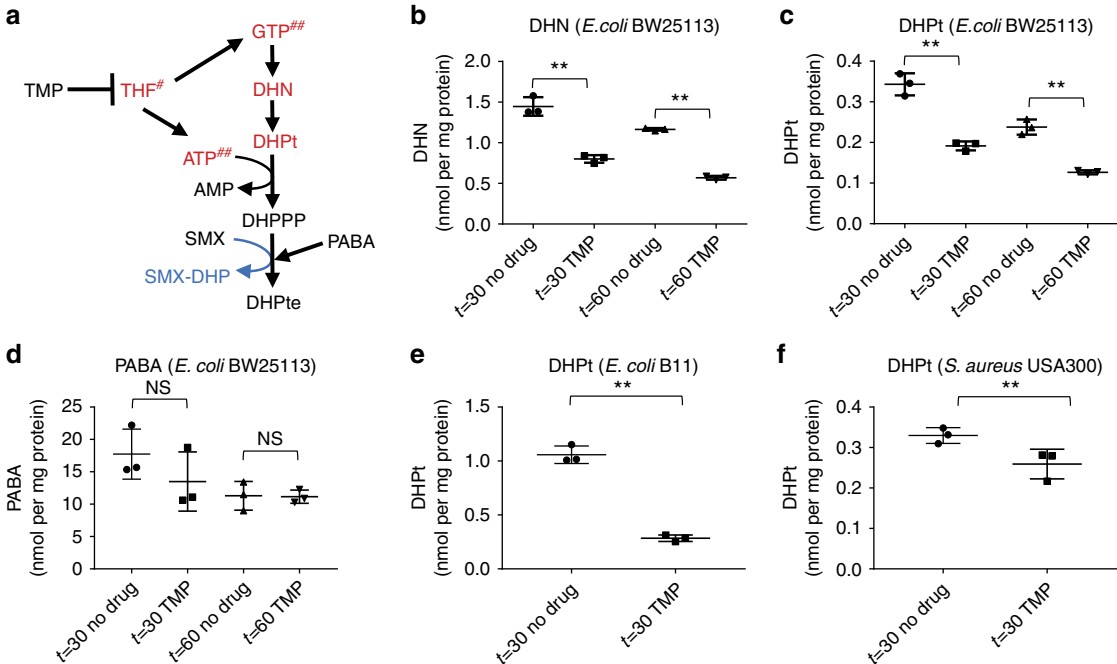

**Fig. 3** TMP inhibits DHPPP biosynthesis pathway. The impact of TMP on intracellular levels of **a** 7,8-dihydroneopterin (DHN) and **b**, **e**, **f** 6-hydroxymethyl-7,8-dihydropterin (DHPt), **c** PABA were determined by using LC-MS/MS. The results represent the mean and standard deviation of three biological replicates. \*\*$p < 0.05$. NS indicates no significant difference ($p > 0.05$). $p$-values of pairwise comparisons were calculated by using the Student's $t$-test. **d** Effects of TMP on DHPPP biosynthesis pathway. Red characters are metabolites that decrease intracellular amount in response to TMP treatment (THF[23], GTP[29], and ATP[29] were reported previously)

ultimately potentiates SMX susceptibility by impairing the DHPPP biosynthesis pathway. Consistent with this hypothesis, we observed that inosine and methionine did not antagonize SMX activity in the Δ*nudB* strain, indicating that these metabolites interfere with SMX action by increasing metabolic flux towards DHPPP biosynthesis (Fig. 2b). Taken together, our findings suggest the existence of a previously unrecognized functional metabolic connection between downstream and upstream components of the bacterial folate biosynthetic pathway that dramatically impacts on SMX susceptibility.

**TMP inhibits DHPPP biosynthesis**. As decreased DHPPP biosynthesis enhances SMX susceptibility, it is likely to be that TMP potentiates SMX activity by negatively impacting the DHPPP biosynthesis pathway (Fig. 3a). To test this hypothesis, we developed liquid chromatography-tandem mass spectrometry (LC-MS/MS) analytical methods using authentic standards to quantify two of the DHPPP precursors, 7,8-dihydroneopterin (DHN) and 6-hydroxymethyl-7,8-dihydropterin (DHPt). We verified that when treated with TMP, *E. coli* BW25113 strain showed significantly lower intracellular levels of both DHN and DHPt, relative to untreated controls (Fig. 3b, c). TMP treatment did not affect intracellular level of PABA, suggesting that the TMP treatment only affected intracellular levels of a specific set of metabolites (Fig. 3d). TMP treatment also negatively affected DHPt levels in *E. coli* B11 and *S. aureus* USA300 strains (Fig. 3e, f). Therefore, inhibition of THF biosynthesis by TMP directly impairs DHPPP biosynthesis.

If impairment of DHPPP synthesis is the mechanistic basis for TMP-mediated potentiation of SMX activity, it follows that TMP-SMX synergy should be abolished in the *E. coli* Δ*nudB* strain. To evaluate TMP-SMX synergy we performed growth assays with wild-type *E. coli* BW25113 and the Δ*nudB* strain in the presence

of each drug at a concentration that by itself led to mild growth impairment (Fig. 4a, b). As anticipated when TMP-SMX were combined, their synergistic action resulted in full growth inhibition of the wild-type strain (Fig. 4a). In striking contrast, the combination of TMP-SMX showed an additive inhibitory effect on growth of the *E. coli* Δ*nudB* strain (Fig. 4b). Furthermore, through the use of checkerboard assays we found that TMP-SMX showed additive effects against the Δ*nudB* strain (Fig. 4c). Thus, synergy between TMP and SMX is contingent upon the ability of TMP to disrupt synthesis of DHPPP (Fig. 4d).

## Discussion

Over the past 50 years, the mechanistic basis for the potently synergistic action between TMP and SMX has been explained by an overly simplistic model. Hitchings first proposed that sulfa drugs such as SMX potentiate the action of TMP through inhibition of DHF accumulation, which enhances the interaction of TMP with its target DHF reductase[22]. Several studies have confirmed key elements of this model[23, 24]. However, this model does not account for the ability of TMP to enhance microbial susceptibility to SMX. Our findings reveal that TMP potentiates SMX activity through the disruption of a previously unrecognized metabolic feedback loop and the cyclic mutual potentiation of these disruptions results in amplified depletion of the essential cofactor THF (Fig. 4d). These findings highlight the importance of metabolic pathway structure in understanding antimicrobial drug interaction and will enable the identification of additional pathways that can be explored for potently synergistic antimicrobial based targeting.

## Methods

**Bacterial strains and growth conditions**. Bacterial strains and plasmids used in this study are listed in Supplementary Table 3. Bacterial strains were grown in Lysogeny Broth (LB, Difco) or M9 minimal medium supplemented with 0.2% (vol:

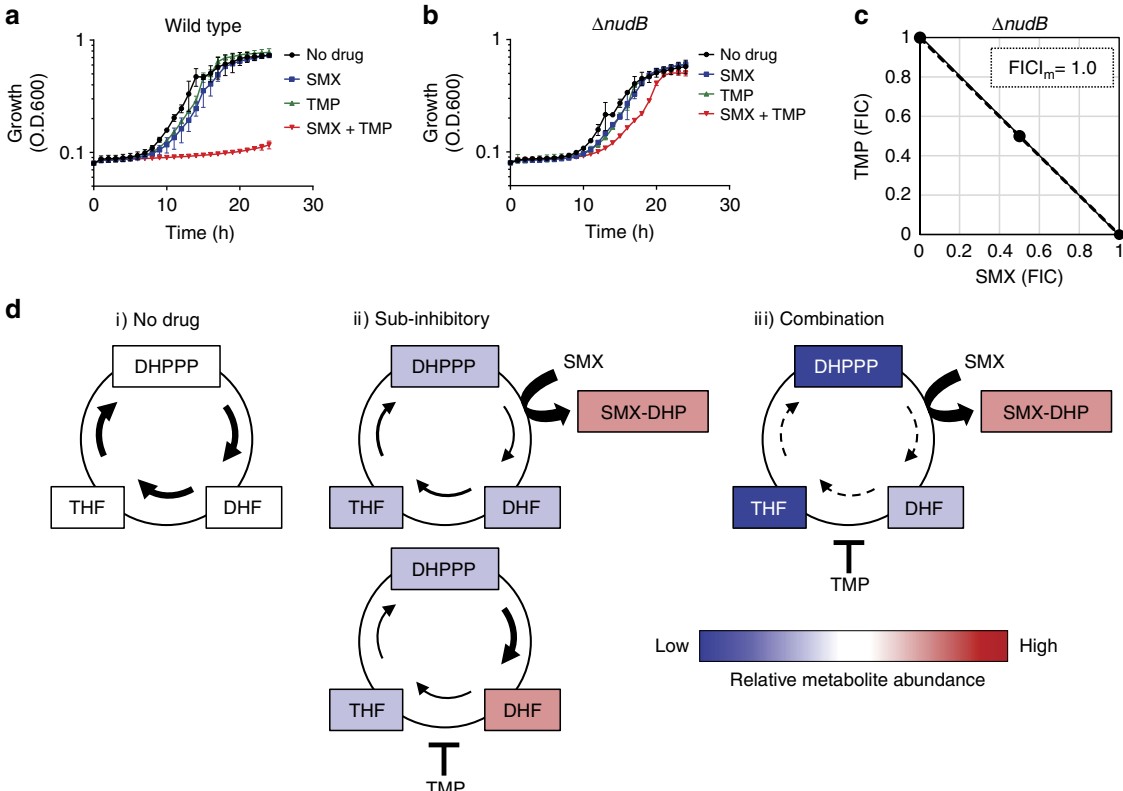

**Fig. 4** A metabolic feedback loop amplifies SMX-TMP activity. **a, b** SMX-TMP synergy was assessed by growth kinetics in the presence of a sub-inhibitory concentration of each drug and in combination. The results represent the mean and SD of three biological replicates. **a** Wild type (*E. coli* BW25113 strain) was grown in the presence of SMX ($0.06\,\mu g\,ml^{-1}$), TMP ($0.125\,\mu g\,ml^{-1}$), or a combination of SMX ($0.06\,\mu g\,ml^{-1}$) and TMP ($0.125\,\mu g\,ml^{-1}$). **b** $\Delta nudB$ (*E. coli* BW25113 $\Delta nudB$) was grown in the presence of SMX ($0.003\,\mu g\,ml^{-1}$), TMP ($0.0125\,\mu g\,ml^{-1}$), or a combination of SMX ($0.003\,\mu g\,ml^{-1}$) and TMP ($0.0125\,\mu g\,ml^{-1}$). **c** SMX-TMP synergy against $\Delta nudB$ (*E. coli* BW25113 $\Delta nudB$) was assessed by FICI. $FICI_m$, minimum value of FICI for the tested combinations is shown. Representative data from at least three independent experiments are shown. **d** Relative metabolite abundance and metabolic flux are shown. No drug treatment (i), sub-inhibitory concentrations of each drug (ii), and combination of sub-inhibitory concentrations of each drug (iii). SMX treatment inhibits the accumulation of DHF by TMP that results in potentiation of TMP activity, and TMP treatment decreases DHPPP that potentiates SMX activity. Cyclic mutual potentiation results in amplified depletion of the essential cofactor THF

vol) glucose (M9-glucose) at 37 °C. 0.5% Casamino acids, pantothenate (f.c. $2\,\mu g\,ml^{-1}$), nicotinamide (f.c. $2\,\mu g\,ml^{-1}$), thiamin (f.c. $2\,\mu g\,ml^{-1}$), and biotin (f.c. $100\,ng\,ml^{-1}$) were supplemented to M9-glucose for growing *S. aureus* USA300 strain. When required, media were amended with penicillin G (Sigma-Aldrich) $150\,\mu g\,ml^{-1}$ or kanamycin (Teknova) $50\,\mu g\,ml^{-1}$.

**Strains construction**. For the complementation analyses, we cloned the *nudB*, *gcvP*, *gcvH*, and *gcvT* genes as follows. The DNA fragment, which contained the respective open reading frame, was amplified by PCR using chromosomal DNA of *E. coli* BW25113 as a template. Primers used for these constructs included *Sac*I and *Bam*HI restriction sites, respectively, and are listed in Supplementary Table 4. The obtained PCR products were digested with *Sac*I and *Bam*HI, gel-purified and then ligated into *Sac*I–*Bam*HI digested pUC19. The resultant plasmid constructs were verified by DNA sequencing.

**Antimicrobial agent susceptibility testing**. SMX and TMP were purchased from Sigma-Aldrich. MAC173979 was synthesized as previously described[17]. The minimum inhibitory concentrations (MICs) of SMX, TMP, and MAC173979 were determined by using the microdilution method.

Twofold dilution series of each antimicrobial agent in M9-glucose was prepared in 96-well round-bottom plates (Corning). Bacterial strains were grown overnight in LB medium, sub-cultured into fresh LB medium, and grown to mid-log phase. The cultures were washed twice and resuspended in M9-glucose, then inoculated into each well containing M9-glucose to $OD_{600}$ 0.001. MICs were determined by visible growth after 24 h incubation at 37 °C. For Supplementary Figure 1, the same drug susceptibility testing was performed in 96 well flat bottom plate (Corning) and MICs were read spectrophotometrically ($OD_{600}$) to determine the minimum amount of antimicrobial agent required to inhibit at least 50% of growth relative to a no drug control after 24 h incubation at 37 °C. Interaction between antimicrobial agents was assessed by using a checkerboard assay format to determine FICI of

each agent in the presence of sub-inhibitory concentrations of another. FICI of antimicrobial agent combinations were calculated using the following equation, FICI = ([MIC drug A in presence of Drug B] [MIC of drug A]$^{-1}$) + ([MIC of drug B in the presence of drug A] [MIC of drug B]$^{-1}$). Minimum FICI ($FICI_m$) represents the lowest fractional combination of antimicrobial agent pairs to achieve growth inhibition. $FICI_m \leq 0.5$ is regarded as synergistic.

**Growth assays**. Bacterial strains were grown overnight in LB medium. The cultures were washed twice and resuspended in M9-glucose then inoculated into each well of a 96-well flat-bottom plate (Corning) containing M9-glucose containing different concentrations of SMX, TMP, or combination of SMX and TMP. Growth was monitored in a Synergy H1 Hybrid Multi-Mode Microplate Reader (BioTek) with the following settings (temperature 37 °C, continuous orbital shaking at 282 cpm (3 mm), read every hour at $OD_{600}$).

**Synthesis of DHPt**. Synthesis of DHPt was accomplished as shown in Supplementary Fig. 3. The synthesis started with the degradation of commercially available folic acid. A solution of folic acid in 40% aqueous hydrogen bromide and excess bromine was heated using a previously described method to obtain 6-formylpterin in 52% yield[25]. Sodium borohydride-mediated reduction of 6-formylpterin in 0.1 N NaOH solution provided 6-hydroxymethylpterin in 95% yield. A well-established method for the selective reduction of 7,8 double bond of folate species was employed for the conversion of 6-hydroxymethylpterin to DHPt. This transformation was accomplished by using sodium dithionite as a reducing agent in the presence of ascorbic acid at pH 6.5[26]. Synthesized DHPt was fully characterized using $^1$H, $^{13}$C NMR and high-resolution mass spectrometry.

**Liquid chromatography-tandem mass spectrometry**
*Sample preparation*: bacterial strains were grown until early log growth phase ($OD_{600}$ 0.2–0.3). The cells were then diluted to $OD_{600}$ 0.2 and were treated with 4

$\mu$g ml$^{-1}$ TMP. After 30 min, and 60 min with or without TMP treatment, 40 ml of each culture was collected by centrifugation. Metabolites were extracted by using 500 $\mu$l of acetonitrile:methanol:water (40:40:20) extraction buffer as previously described[27]. One hundred and fifty microliters of each sample was evaporated using a SpeedVac Concentrator (Savant SC210A, Thermo Scientific) and reconstituted with 50 $\mu$L of 10 mM ammonium acetate (95:5 H$_2$O:MeCN, pH 9.0) solution containing 200 nM internal standard (hydroxypteroic acid) for DHPt and DHN quantification. Similarly, for PABA quantification, 150 $\mu$L of each sample was evaporated and reconstituted with 50 $\mu$L of 10 mM aqueous ammonium acetate (pH 4.0) solution containing 100 nM internal standard (d$_4$-PABA).

*DHPt and DHN quantification*: LC-MS/MS experiments were performed with Shimadzu UFLC-XR (LC) and AB Sciex QTRAP 5500 (MS) instruments. Synthetic DHPt and DHN (Santa Cruz Biotechnology) were used as authentic standards. Hydroxypteroic acid was used as an internal standard. Reverse-phase LC was performed on an Eclipse XDB-C8 column (4.6 × 150 mm, 5 $\mu$m particle size; Agilent, Santa Clara, CA). Mobile phase A was 10 mM ammonium acetate in H$_2$O (pH 9.0), whereas mobile phase B was 10 mM ammonium acetate in 5:95 H$_2$O: MeCN (pH 9.0). Initial conditions were 5% B from 0 to 1.0 min, after which the %B was increased to 70% from 1.0 to 5.3 min, then %B was increased to 90% from 5.3 to 5.7 min. The column was washed with 90% B from 5.7 to 6.5 min, returned to 5% B at 7.0 min, and allowed to re-equilibrate for 4 min in 5% B, to provide a total run time of 11 min. The flow rate was 0.5 mL min$^{-1}$ and the column oven was maintained at 35 °C. The injection volume was 10 $\mu$L. All analytes were analyzed by MS in positive ionization mode by Multiple Reaction Monitoring (MRM). To determine the optimum MRM settings (Supplementary Table 3), each analyte was infused at a concentration of 10 $\mu$M in 50:50 H$_2$O:MeCN containing 10 mM ammonium acetate at pH 9.0 onto the MS by a syringe pump at a flow of 10 $\mu$L min$^{-1}$. During direct infusion, we also observed the presence of oxidized forms of both the analytes, i.e., the presence of neopterin and 6-hydroxymethylpterin. Such auto air oxidation of reduced form of pterin species has been known; thus, we decided to measure both the oxidized and reduced forms of these species (Supplementary Table 3). Peak areas of both species were combined for the analytes before converting to the concentration. Analyte and internal standard peak areas were calculated (MultiQuant, version 2.0.2). Analyte peak areas were normalized to internal standard peak areas and the concentrations of DHN and DHPt were determined using appropriate standard curves.

*PABA quantification*: PABA was quantified using a previously developed LC-MS/MS method[28]. d$_4$-PABA was used as an internal standard. Reverse-phase LC was performed on a Kinetix C18 column (50 × 2.1 mm, 2.6 $\mu$m particle size; Phenomenex, Torrance, CA). Mobile phase A was 10 mM ammonium acetate in H$_2$O (pH 4.0), whereas mobile phase B was 0.1% formic acid in 5:95 H$_2$O:MeOH. Initial conditions were 5% B, after which the proportion of B was increased to 10% in 1.0 min, 15% in 4.0 min, 55% in 4.5 min, 92% in 4.75 min, returned to initial composition of eluent (5% B) in 5.0 min, and then held for 3.0 min in order to re-equilibrate the column, which provided a total run time of 8 min. The flow rate was 0.5 mL min$^{-1}$ and the column oven was maintained at 35 °C. The injection volume was 10 $\mu$L. Both analyte and internal standard were analyzed by electrospray ionization MS in positive ionization mode by MRM. The transformations $m/z$ 138.05 → 94.07 for PABA and 142.08 → 98.09 for d$_4$-PABA were used for MRM (Supplementary Table 5).

**Data availability**. All relevant data are available in this article and its Supplementary Information files, or from the corresponding authors upon request.

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

## Acknowledgements

We thank Peter Southern for critical reading of the manuscript and helpful comments, Bruce Witthuhn for expert technical assistance with LC-MS/MS analysis, Arkady B. Khodursky for providing us the *E. coli* single-gene deletion mutants, Ryan C. Hunter for providing *S. aureus* USA300, and Betsy Hirsch for providing *E. coli* B11. We also thank Allison Bauman for her excellent technical assistance. This work was supported by a grant from the University of Minnesota Academic Health Center Faculty Research Development Program to A.D.B and C.C.A., and by startup funds from the University of Minnesota to A.D.B. and C.C.A.

## Author contribution

Y.M., A.B., S.L.K., and S.D. performed experiments. S.D. synthesized custom reagents. Y. M., C.C.A., and A.D.B. conceived the work. Y.M., S.D., and A.D.B. wrote the manuscript. All authors contributed to analyzing data and editing of the manuscript.

**Additional information**

**Competing interests:** The authors declare no competing interests.

