## [Peer Review File · Nature Communications]

Reviewers' comments:

Reviewer #1 (Remarks to the Author):

This is an excellent study that re-examines the mechanistic basis of the synergistic interaction between sulfonamides (SMX) and trimethoprim (TMP) antibacterial agents. Through detailed mechanistic studies, the authors describe the discovery of a novel mutual potentiation mechanism of each drug on each other. The experiments are well planned and easy to follow, producing high confidence in their results, including the key finding that synergy is contingent upon the ability of TMP to disrupt the synthesis of DHPPP.

This manuscript has the potential to highly significant, and consequently, it is recommended that the authors provide some additional data to round out this study upon revision.

Suggested revisions and general comments:

Line 33. Add the chemical name of MAC

Figure 1. Legend lacks sufficient experimental details to evaluate the figure. Bacterial strain utilized, media and drug concentration ranges utilized should be provided

MAC137979 is a dually electrophilic species, which is expected to exhibit polypharmacology when used at higher concentrations, care should be taken to over interpret mechanistic & MIC results from this molecule.

The FICI method used in this study is a simplistic assessment method for synergy and antagonism that has statistical limitations (see Scientific Reports 6, Article number: 25523 (2016) doi:10.1038/srep25523). One critical assumption is that the drug response curves are identical between the drugs utilized in the study. Raw checkerboard data (images) should be included in the supplementary data section, to allow others to examine the data. If the growth inhibition curve of SMX, TMP or MAC is significantly different (i.e. if one drug trails) FICI method should not be used in this manuscript.

Line 50. Reference 5, it is recommended to change the first citation to Science 2012: Vol. 335, Issue 6072, pp. 1110-1114 which fully describes the biochemical mechanism of DHPS.

The authors should confirm that the NudB mutant strain does not have other non-mechanistic drug sensitivities to unrelated antibiotics (Fluoroquinolones, nitrofurantoin etc.)

Figure 3 A & B, lacks an internal control metabolite (pABA?) to demonstrate that this is a specific effect and not related to general growth inhibition in the drug treated vs control.

It has been our experience when studying synergy and antagonism amongst other antibiotic classes that sometimes observed events turn out to be strain specific, due to the individual genetic backgrounds of lab adapted strains. It is recommended that the authors demonstrate that the proposed mutual potentiation mechanism between TMP and SMX, is observable in multiple strains and ideally bacterial species, which would undoubtedly boost the significance of the study.

Reviewer #2 (Remarks to the Author):

In this manuscript, the authors revisit the widely assumed mechanism of the synergy between trimethoprim (TMP) and sulfamethoxazole (SMX). This work is important in part because these drugs are currently widely used for bacterial infections, but also because understanding the mechanism of antibiotic classes in general enable improvement in our approach for design for new antibiotics. Due to the ever growing problem of antibiotic resistance, new drugs are needed. The same understanding can find application in the search for drugs combating infections by other organisms. Since folate biosynthetic pathways exist only in bacteria (and plants) it is a useful pathway to target.

TMP and SMX are the drugs that comprise the commonly used antibiotic-combination drug Bactrim. Both drugs impact enzymes involved in bacterial synthesis of the vitamin folic acid, with somewhat different mechanisms. Sulfa drugs target dihydropterate synthase, and share structural similarities to substrate p-aminobenzoic acid; the sulfa drugs actually serve as alternative substrates which generate intermediates that cannot advance in the pathway (and 'use up' the precursor). TMP binds and inhibit dihydrofolate reductase, a later step in the pathway. It is widely accepted that SMX potentiates the action of TMP because SMX acts earlier in the pathway, and limits production of dihydrofolate. The authors cite published mathematical models that support this idea. However, current thought does not offer a mechanism for the potent synergism that TMP also has on SMX.

Here, the authors first demonstrate the synergism each drug has on the other, then systematically perform experiments that detail how this synergism works, using laboratory strains of *E. coli*. The authors used three compounds that target steps in the folate pathway, including TMP, SMX, and additionally "MAC" a compound that inhibits one of the two steps between aromatic precursor chorismate and PABA. Checkerboard assays demonstrated that inhibitors earlier in the pathway demonstrated potentiation with inhibitors targeting later step; their data confirmed the observation that there was a very potent synergism between TMP and SMX, well beyond what was observed between pairs of the other compounds, including MAC and TMP, or MAC and SMX. Next, MIC experiments were performed with a set of single gene knock-out mutants which lacked specific enzymes in various steps in folate metabolism and folate biosynthesis; MIC measurements were used to identify particularly susceptible steps. An *E. coli* mutant strain lacking nudB (an upstream step between GTP and dihydropteroate synthase) showed vastly increased susceptibility to both SMX and TMP, consistent with the idea that the pterin branch of the folate pathway is more limiting for folate synthesis than the PABA branch. Incubation of wild type cells with TMP resulted in decreased levels of two pterin pathway intermediates, confirming a connection between TMP and the early pterin pathway. Next the authors disrupted two genes involved in generation of 10-formyl-tetrahydrofolate and used MIC measurements again to demonstrate a connection between folate interconversions and the early pterin pathway. Finally, they found that inosine and methionine together erased the susceptibility of the nudB knockout to SMX.

In summary, the authors identified an important metabolic link between GTP and folate metabolism that has gone unappreciated but impacts SMX/TMP sensitivity, and used a variety of techniques to support their hypothesis. This work is clearly written, nicely laid out, and the authors make a compelling case for their conclusions. The experiments are well-done, logical, and important.

I do have a few suggestions/comments:

1. The authors state in lines 49-50 that "it was recently demonstrated that SMX acts by competing with PABA for ligation with DHPPP" and cite a 2014 paper. The fundamental concept that sulfa drugs compete with PABA for site on this enzyme, and also that this enzyme catalyzes the incorporation of the sulfa drug into a 'dead end' molecule that cannot function, was demonstrated by the laboratory of

Gene Brown in the 1960's. (Brown, G. 1962 "The Biosynthesis of Folic Acid. II. Inhibition by sulfonamides. J. Biol.Chem. 237:2, 536-540.) It always gives me a pang of sadness when the seminal work of earlier scientists goes unrecognized when we all "stand on the shoulder of giants." (Newton, Bernard of Chartres)

2. The bar graph figures (2C, 3A, 3B) are difficult to interpret given the small size of the graphs and the design of the key. I would recommend using solid black and white as fill in addition to several versions of simple hashmarks or checkerboard design. Color bars would be even better.

3. It is not particularly surprising to me that supplementing cells with the purine base inosine (precursor to both GTP and ATP) and methionine can suppress the sensitivity of cells to anti-folate drugs. Methionine and purine biosynthesis are two major products stemming from enzymes that use folate as carriers of one-carbon units, so presumably less folate metabolism is needed when end products are supplied. The authors might find the following paper helpful as a reference for their work particularly with regard to their studies with glycine cleavage and serine hydroxymethyltransferase mutants. [Dev and Harvey (1982) "Sources of one-carbon units in the folate pathway of Escherichia coli" J. Biol. Chem. 257:4, pp 1980-1986].

4. The final question that comes to my mind is whether the levels of GTP were directly impacted by the combination of SMX/TMP.

Response to Referees

We sincerely thank the reviewers for their careful and thoughtful evaluation of our manuscript. We have made the suggested additions and modifications, and we feel that these changes have improved the overall quality of the manuscript. These changes are highlighted in the updated manuscript files.

Reviewer #1 (Remarks to the Author):

This is an excellent study that re-examines the mechanistic basis of the synergistic interaction between sulfonamides (SMX) and trimethoprim (TMP) antibacterial agents. Through detailed mechanistic studies, the authors describe the discovery of a novel mutual potentiation mechanism of each drug on each other. The experiments are well planned and easy to follow, producing high confidence in their results, including the key finding that synergy is contingent upon the ability of TMP to disrupt the synthesis of DHPMP.

This manuscript has the potential to highly significant, and consequently, it is recommended that the authors provide some additional data to round out this study upon revision.

Suggested revisions and general comments:

Line 33. Add the chemical name of MAC

Response: The chemical name of MAC173979, 3-dichloro-1-(3-nitrophenyl)prop-2-en-1-one, was added to the manuscript (Line 34-35).

Figure 1. Legend lacks sufficient experimental details to evaluate the figure. Bacterial strain utilized, media and drug concentration ranges utilized should be provided

Response: Bacterial strains, growth medium and drug concentration ranges utilized were added to Figure 1 legend.

MAC137979 is a dually electrophilic species, which is expected to exhibit polypharmacology when used at higher concentrations, care should be taken to over interpret mechanistic & MIC results from this molecule.

Response: We totally agreed that MAC137979 contains a Michael acceptor that may react non-specifically with other proteins, especially at high concentrations. However, near it's MIC, MAC173979 appears to act fairly specifically because its activity can be fully rescued by exogenous PABA indicating it acts quite specifically on PABA biosynthesis. Eric Brown has reported that MAC173979 is an irreversible time-dependent inhibitor of PABA biosynthesis using a reconstituted enzyme assay with the three enzyme system PabA-PabB-PabC with a K_i of 7 μM , which is very close to its MIC of 8 μM (2 $\mu\text{g/mL}$). We assessed synergy at MAC173979 concentrations below 6 $\mu\text{g/mL}$. Thus, we think the majority of the activity can be attributed to on-target activity.

The FICI method used in this study is a simplistic assessment method for synergy and antagonism that has statistical limitations (see Scientific Reports 6, Article number: 25523 (2016) doi:10.1038/srep25523). One critical assumption is that the drug response curves are identical between the drugs utilized in the study. Raw checkerboard data (images) should be

included in the supplementary data section, to allow others to examine the data. If the growth inhibition curve of SMX, TMP or MAC is significantly different (i.e. if one drug trails) FICI method should not be used in this manuscript.

Response: We had difficulty producing quality raw images of our checkerboard assays. Therefore, we repeated our assays using flat bottom microtiter plates (Corning) and optical density (600 nm) was measured after 24 hrs at 37 °C using a Synergy H1 Hybrid Multi-Mode Microplate Reader (BioTek). These results are consistent with those presented in Figure 1. We have included these data in the supplementary material as suggested (Fig. S1).

Line 50. Reference 5, it is recommended to change the first citation to Science 2012: Vol. 335, Issue 6072, pp. 1110-1114 which fully describes the biochemical mechanism of DHPS.

Response: Reference 5 has been changed as suggested (now ref 8). This reference was in the original draft and was inadvertently deleted during manuscript revision.

The authors should confirm that the NudB mutant strain does not have other non-mechanistic drug sensitivities to unrelated antibiotics (Fluoroquinolones, nitrofurantoin etc.)

Response: We tested two other unrelated antibiotics (ceftazidime and ciprofloxacin) and found that susceptibility of the *nudB* mutant strain was unchanged relative to the parental strain. The results were added as Table S4 and to the main text (Lines 71-74).

Figure 3 A & B, lacks an internal control metabolite (pABA?) to demonstrate that this is a specific effect and not related to general growth inhibition in the drug treated vs control.

Response: As suggested, we used PABA as an internal control metabolite and abundance was found to be stable during TMP treatment. The results were added to the text (Lines 113-115) and as Fig. 3 C.

It has been our experience when studying synergy and antagonism amongst other antibiotic classes that sometimes observed events turn out to be strain specific, due to the individual genetic backgrounds of lab adapted strains. It is recommended that the authors demonstrate that the proposed mutual potentiation mechanism between TMP and SMX, is observable in multiple strains and ideally bacterial species, which would undoubtedly boost the significance of the study.

Response: We appreciate this suggestion. We used two additional strains, *E.coli* B11 and *S. aureus* USA300. Both strains are clinical isolates. SMX-TMP combinations were synergistically active against these strains while MAC173979-TMP combination did not show synergistic activity against these strains. TMP treatment also decreased intracellular DHPt levels of these strains suggesting TMP negatively affects DHPPP synthesis in these strains. Together, these results demonstrated that the proposed mutual potentiation mechanism between TMP and SMX is robust and observable in different strains of *E.coli* and *S. aureus*. These results were added to the main text (Line 47-50, Line115-116), Fig. 1 E, 3E, and 3F.

Reviewer #2 (Remarks to the Author):

In this manuscript, the authors revisit the widely assumed mechanism of the synergy between

trimethoprim (TMP) and sulfamethoxazole (SMX). This work is important in part because these drugs are currently widely used for bacterial infections, but also because understanding the mechanism of antibiotic classes in general enable improvement in our approach for design for new antibiotics. Due to the ever growing problem of antibiotic resistance, new drugs are needed. The same understanding can find application in the search for drugs combating infections by other organisms. Since folate biosynthetic pathways exist only in bacteria (and plants) it is a useful pathway to target.

TMP and SMX are the drugs that comprise the commonly used antibiotic-combination drug Bactrim. Both drugs impact enzymes involved in bacterial synthesis of the vitamin folic acid, with somewhat different mechanisms. Sulfa drugs target dihydropterate synthase, and share structural similarities to substrate p-aminobenzoic acid; the sulfa drugs actually serve as alternative substrates which generate intermediates that cannot advance in the pathway (and 'use up' the precursor). TMP binds and inhibit dihydrofolate reductase, a later step in the pathway. It is widely accepted that SMX potentiates the action of TMP because SMX acts earlier in the pathway, and limits production of dihydrofolate. The authors cite published mathematical models that support this idea. However, current thought does not offer a mechanism for the potent synergism that TMP also has on SMX.

Here, the authors first demonstrate the synergism each drug has on the other, then systematically perform experiments that detail how this synergism works, using laboratory strains of *E. coli*. The authors used three compounds that target steps in the folate pathway, including TMP, SMX, and additionally "MAC" a compound that inhibits one of the two steps between aromatic precursor chorismate and PABA. Checkerboard assays demonstrated that inhibitors earlier in the pathway demonstrated potentiation with inhibitors targeting later step; their data confirmed the observation that there was a very potent synergism between TMP and SMX, well beyond what was observed between pairs of the other compounds, including MAC and TMP, or MAC and SMX. Next, MIC experiments were performed with a set of single gene knock-out mutants which lacked specific enzymes in various steps in folate metabolism and folate biosynthesis; MIC measurements were used to identify particularly susceptible steps. An *E. coli* mutant strain lacking nudB (an upstream step between GTP and dihydropterate synthase) showed vastly increased susceptibility to both SMX and TMP, consistent with the idea that the pterin branch of the folate pathway is more limiting for folate synthesis than the PABA branch. Incubation of wild type cells with TMP resulted in decreased levels of two pterin pathway intermediates, confirming a connection between TMP and the early pterin pathway. Next the authors disrupted two genes involved in generation of 10-formyl-tetrahydrofolate and used MIC measurements again to demonstrate a connection between folate interconversions and the early pterin pathway. Finally, they found that inosine and methionine together erased the susceptibility of the nudB knockout to SMX.

In summary, the authors identified an important metabolic link between GTP and folate metabolism that has gone unappreciated but impacts SMX/TMP sensitivity, and used a variety of techniques to support their hypothesis. This work is clearly written, nicely laid out, and the authors make a compelling case for their conclusions. The experiments are well-done, logical, and important.

I do have a few suggestions/comments:

1. The authors state in lines 49-50 that "it was recently demonstrated that SMX acts by competing with PABA for ligation with DHPPP" and cite a 2014 paper. The fundamental concept that sulfa drugs compete with PABA for site on this enzyme, and also that this enzyme

catalyzes the incorporation of the sulfa drug into a 'dead end' molecule that cannot function, was demonstrated by the laboratory of Gene Brown in the 1960's. (Brown, G. 1962 "The Biosynthesis of Folic Acid. II. Inhibition by sulfonamides. J. Biol.Chem. 237:2, 536-540.) It always gives me a pang of sadness when the seminal work of earlier scientists goes unrecognized when we all "stand on the shoulder of giants." (Newton, Bernard of Chartres)

Response: Thank you very much for this information! We added the suggested citation as reference 7 and modified the main text (Line 55-58).

2. The bar graph figures (2C, 3A, 3B) are difficult to interpret given the small size of the graphs and the design of the key. I would recommend using solid black and white as fill in addition to several versions of simple hashmarks or checkerboard design. Color bars would be even better.

Response: We have modified Fig. 2C, 3A, and 3B as suggested.

3. It is not particularly surprising to me that supplementing cells with the purine base inosine (precursor to both GTP and ATP) and methionine can suppress the sensitivity of cells to anti-folate drugs. Methionine and purine biosynthesis are two major products stemming from enzymes that use folate as carriers of one-carbon units, so presumably less folate metabolism is needed when end products are supplied. The authors might find the following paper helpful as a reference for their work particularly with regard to their studies with glycine cleavage and serine hydroxymethyltransferase mutants. [Dev and Harvey (1982) "Sources of one-carbon units in the folate pathway of *Escherichia coli*" J. Biol. Chem. 257:4, pp 1980-1986].

Response: We agree that one might anticipate that supplementation of cultures with inosine and methionine could suppress the susceptibility of cells to all anti-folate drugs. However, inosine and methionine selectively antagonize the activity of SMX (and other sulfa drugs) but not TMP (Line 96-97, Fig. S2). Our data suggest that inosine-mediated antagonism of SMX is in part due to its ability to sustain flux to DHPPP (overcomes metabolic wasting), and TMP is not antagonized because its activity is not impacted by increased availability of DHPPP. We are actively exploring this mechanism of antagonism and plan to report on our findings in the coming months.

4. The final question that comes to my mind is whether the levels of GTP were directly impacted by the combination of SMX/TMP.

Response: Kwon, Higgins and Rabinowitz previously demonstrated that SMX/TMP treatment impacts GTP levels in *E. coli* (Kwon *et al* 2010 ACS Chem Biol; ref 21 of the manuscript). This observation is reflected in our schematic in Fig. 3D (previously Fig. 3C).

REVIEWERS' COMMENTS:

Reviewer #1 (Remarks to the Author):

The author has done an excellent job of responding to the reviewer comments.

REVIEWERS' COMMENTS:

Reviewer #1 (Remarks to the Author):

The author has done an excellent job of responding to the reviewer comments.

Author response:

Thank you.